# Diabetic Retinopathy Environment-Wide Association Study (EWAS) in NHANES 2005–2008

**DOI:** 10.3390/jcm9113643

**Published:** 2020-11-12

**Authors:** Kevin Blighe, Sarega Gurudas, Ying Lee, Sobha Sivaprasad

**Affiliations:** 1Institute of Ophthalmology, University College London, London EC1V 9EL, UK; k.blighe@ucl.ac.uk (K.B.); sarega.gurudas.17@ucl.ac.uk (S.G.); ying.lee@tu-dresden.de (Y.L.); 2National Institute for Health Research, Moorfields Biomedical Research Centre, London EC1V 2PD, UK

**Keywords:** environment wide association study, diabetic retinopathy, hyperglycaemia, circulating biomarkers

## Abstract

Several circulating biomarkers are reported to be associated with diabetic retinopathy (DR). However, their relative contributions to DR compared to known risk factors, such as hyperglycaemia, hypertension, and hyperlipidaemia, remain unclear. In this data driven study, we used novel models to evaluate the associations of over 400 laboratory parameters with DR compared to the established risk factors. Methods: we performed an environment-wide association study (EWAS) of laboratory parameters available in National Health and Nutrition Examination Survey (NHANES) 2007–2008 in individuals with diabetes with DR as the outcome (test set). We employed independent variable (feature) selection approaches, including parallelised univariate regression modelling, Principal Component Analysis (PCA), penalised regression, and RandomForest™. These models were replicated in NHANES 2005–2006 (replication set). Our test and replication sets consisted of 1025 and 637 individuals with available DR status and laboratory data respectively. Glycohemoglobin (HbA1c) was the strongest risk factor for DR. Our PCA-based approach produced a model that incorporated 18 principal components (PCs) that had an Area under the Curve (AUC) 0.796 (95% CI 0.761–0.832), while penalised regression identified a 9-feature model with 78.51% accuracy and AUC 0.74 (95% CI 0.72–0.77). RandomForest™ identified a 31-feature model with 78.4% accuracy and AUC 0.71 (95% CI 0.65–0.77). On grouping the selected variables in our RandomForest™, hyperglycaemia alone achieved AUC 0.72 (95% CI 0.68–0.76). The AUC increased to 0.84 (95% CI 0.78–0.9) when the model also included hypertension, hypercholesterolemia, haematocrit, renal, and liver function tests.

## 1. Introduction

Diabetic retinopathy (DR), a chronic diabetes complication, is generally believed to be the most common cause of microvascular changes in the retina. The initial retinal lesions of diabetic retinopathy (DR) are microaneurysms, but they can occur in eyes with and without diabetes [1,2,3]. With increasing duration of diabetes, other lesions develop and co-exist in the retina, such as retinal haemorrhages, exudates, intraretinal microvascular abnormalities, and neovascularization of the retina or optic disc. Based on the presence of individual lesions or a constellation of them, DR severity level is graded from mild, moderate, and severe non-proliferative diabetic retinopathy (NPDR) to proliferative diabetic retinopathy (PDR) [4,5]. Diabetic macular oedema (DME) can occur in any stage of DR [5]. In population-based studies, approximately a third of people with diabetes have DR [6,7].

The established systemic risk factors for DR are suboptimal control of hyperglycaemia, hypertension, and hyperlipidaemia [8,9]. Hypertension can also cause some of these retinal lesions independent of diabetes [10]. There are several laboratory parameters that have been shown to be abnormal in people with DR, such as hyperuricemia [11], low vitamin D levels [12], low thyroxine levels [13], anaemia [14], oxidative stress, and inflammatory markers [15]. In addition, DR is also associated with markers of diabetic kidney disease, including microalbuminuria and serum creatinine [16,17], and cardiovascular disease markers, such as raised C-reactive protein (CRP) [18]. Most of these associations and risks of DR are reported based on analysis of candidate laboratory-based serum or urinary markers.

In addition to these risk factors, there are several other non-modifiable and modifiable risk factors that have been attributed to the development and progression of DR. Some of these include age of onset of diabetes, duration of diabetes, male sex, and ethnicity [19,20,21]. However, the relative associations of these risk factors and circulating biomarkers with DR is unclear.

There is an unmet need to rank these reported retinal, systemic (blood pressure, body mass index) risk factors, and laboratory markers so that preventative measures may systematically target modifiable factors and markers. As DR is a multifactorial complication of diabetes involving several metabolic and biochemical pathways, ranking will inform clinical practice on how to systematically evaluate and control appropriate risk factors or biomarkers based on a rank order. The ranking may also identify new druggable targets.

The National Health and Nutrition Examination Survey (NHANES—https://wwwn.cdc.gov/Nchs/Nhanes/) was initiated in the 1960s in order to examine the health and nutritional status of U.S. citizens. Since 1999, it has examined approximately 5000 citizens per year, and the data includes diabetes, eye diseases, and several laboratory markers, including environmental toxins, allergens, and pollutants.

In this study, we used an environment wide association study (EWAS) methodology [22,23,24,25] on NHANES 2007–2008 to evaluate the rank order of systemic and laboratory risks of DR among individuals identified as having diabetes to evaluate their relative associations with DR. Our findings were then replicated in NHANES 2005–2006. Our objective is not to only use previously reported risk factors, but also provide new research avenues from this data-driven model-agnostic study.

## 2. Methods

### 2.1. Study Data Collection

We used National Health and Nutrition Examination Survey (NHANES) 2007–2008 as our primary cohort and 2005-2006 as a replication cohort. Both datasets were prepared in the same fashion; however, for ease of interpretation, the following methods describe 2007–2008.

### 2.2. Data Collection

Specifically, three main categories of data were used: examination data (Ophthalmology—Retinal Imaging data), demographics data, and laboratory data. Demographic data included age, sex, and ethnic origin. Socioeconomic variables included education, marital status, and family income.

### 2.3. Diabetes Status

Definition of diabetes: the selection criteria for diabetes included self-reported diabetes, on anti-diabetes drugs, taking insulin, fasting blood sugar (FBS) ≥ 6.1 (110 mg/dL), random blood sugar (RBS) ≥ 11.1 (200 mg/dL), oral glucose tolerance test (OGTT) ≥ 200 mg/dL, glycohemoglobin (HbA1c) ≥ 6.5%.

### 2.4. Diabetic Retinopathy

The NHANES cohort underwent fundus photography using the Canon CR6-45NM ophthalmic digital imaging system and Canon EOS 10D digital camera (Canon, Tokyo, Japan). Two digital images per eye were captured through a non-pharmacologically dilated pupil, one centred on the macula and the second on the optic nerve. The digital images were graded by masked photo graders at the University of Wisconsin Ocular Epidemiologic Reading Center, Madison, using a modification of the Airlie House classification system. The main outcome of interest in the examination data was “4 levels retinopathy severity, worse eye”—this variable was recoded as binary with levels: no retinopathy; retinopathy (including mild non-proliferative retinopathy (NPR), moderate/severe NPR, and proliferative).

### 2.5. Covariates

Age, ethnicity, and diabetes duration were used as covariates. Diabetes duration was calculated as age at screening, minus the age at which the individual was first informed that he/she had diabetes.

### 2.6. Statistical Analysis

Prior to any analysis, in addition, any variable that contained a single value occupying more than 90% of total values was removed, as were variables that had more than 90% missingness. Further specific filtering and encoding was then applied per dataset. (A) Examination data: variables that were different encodings of the main outcome were removed; variables related to the status of the examination appointment were removed; variables related to glaucoma, for which there is already a single variable, were removed; variables related to the left or right eye where there was already a variable for ‘worse’ eye were removed; values encoded as missing were recoded as not available; and all other remaining variables were encoded as binary, with 0 representing the absence of the condition, and 1 representing any recorded presence (at any level) of the condition. (B) Demographic data: variables associated with interpreters and the language of the interview were removed; variables that were duplicates or different encoding of each were removed. (C) Laboratory data: categorical variables were removed and only continuous retained; duplicate variables related to the oral glucose tolerance test were removed; variables related to time since domestic activities (“pump gas”, “shower”, etc.) were removed; variables that were duplicates or different encodings of each were removed; variables measured on the imperial system of weights and measures were removed if they had a corresponding variable in SI units. We focused only on continuous laboratory variables for the following reasons: 1, in NHANES, the majority of categorical variables are derived from the continuous variables; 2, our Principal Component Analysis (PCA)-based approach can only work on continuous variables; 3, for RandomForest™, having continuous variables increases the number of splitting points in the data, and metrics of importance such as Gini are known to exhibit less bias on such data [26].

All datasets were downloaded as SAS XPORT (xpt) format and read into R (v4.0.2) via the *Hmisc* package.

Individuals with a missing value in the main outcome variable were removed before aligning the examination, demographics, and laboratory data via each individual’s respondent sequence number (SEQN). This dataset was then further filtered for only those individuals who had diabetes (Figure 1). Variables were removed from the data that had 0 variance (i.e., constant values) (Appendix A).

Prior to statistical analysis, continuous laboratory variables were logged (log_e_) and then transformed into z-scores to ensure that these were on the same scale. In regression analysis, the complex sampling design of the NHANES dataset was accounted for through use of survey sampling weights via the *survey* package in R/CRAN. To do this, the following value-pairs were used with the *svydesign* function: (*id*, SDMVPSU; *strata*, SDMVSTRA; *weights*, WTMEC2YR; *nest*, TRUE).

Univariate analysis was performed on all candidate predictors using a survey-weighted compute-parallelized logistic regression model via the R/Bioconductor package *RegParallel*, adjusting for age, ethnicity, and duration of diabetes separately. The Benjamini–Hochberg [27] procedure was used to control the type I error false discovery rate (FDR). A customized Manhattan plot was generated using *ggplot2*, while pairwise scatter and correlation plots were generated via a customized pairs plot. Finally, a heatmap was generated via the R/Bioconductor package *ComplexHeatmap*.

As our study is also hypothesis-generating, multivariate approaches based on principal component analysis (PCA), penalised regression, and the RandomForest™ classification algorithm were additionally used. Variables were pre-filtered and prepared as per univariate testing. Principal component analysis was performed via the R/Bioconductor package *PCAtools*. After conducting PCA, each eigenvector was then independently regressed against retinopathy outcome via binary logistic regression and those that passed *p* ≤ 0.05 were used to construct a multivariable model that was further tested in ROC analysis via the *pROC* package in R.

Separately, as model complexity and multi-collinearity can arise from a large number of predictors, elastic net regularization (penalised regression with L1 and L2 penalties of the Lasso and Ridge methods) was used to reduce the number of predictor variables using *glmnet* in R/CRAN. To fit the model, 100x cross-validation was used and alpha (α) set to 0.5. The final chosen variables were those whose coefficients were not shrunk to zero—these were plot as violin plots with scatter overlays to show differences between non-DR and DR via *ggplot2*. To determine accuracy, model predictions were made on the data using the lambda (λ) one-standard-error rule using the *predict* function from the *stats* package in R.

Finally, the RandomForest™ (RF) model was fitted via the *randomForest* R/CRAN package. For this, the dataset was divided randomly into 50% training and 50% validation. Prior to model fitting, the initial model was tuned using functionality provided by the *caret* package in R/CRAN, as follows: (1), a 10x cross-validation control function was defined via *trainControl* function; (2) the best value for ‘mtry’, i.e., the ideal number of variables to randomly sample, was determined using the *train* function across a search/tuning grid ranging between 1–40 and with Kappa as the metric; and (3) using the selected value of ‘mtry’, the ideal number of trees, ‘ntrees’, was determined also via the *train* function with selection metric based on Kappa. After the initial model was fit, variables with mean decrease in accuracy ≤1% were excluded and the model re-fit. This was then repeated in a recursive fashion until all variables with negative mean decrease accuracy were removed from the model.

### 2.7. Final Risk Models

Variables selected from RandomForest™ were grouped based on similarity of function or clinical use. Each group was then used to create independent univariate or multivariable binary logistic regression models with DR as the end-point. A single Wald test *p*-value was derived for each model using *wald.test* from the *aod* package. ROC analysis was performed using *pROC*. McFadden’s and Nagelkerke’s pseudo-R^2^ were derived via the *pscl* and *rms* packages, respectively.

## 3. Results

### 3.1. Study Cohort

In NHANES 2007–2008, retinal imaging data is available for 3863 individuals, demographics data is available for 10,149 individuals, and laboratory data is available for between 394 and 9307 individuals, depending on the individual laboratory dataset in NHANES (see Figure 1 footnote). After aligning all data and filtering for those who had diabetes by our classification, 1025 individuals remained in our dataset. The selection process is illustrated in Figure 1, while Table 1 provides an overview of the demographics of these individuals.

For our replication cohort, NHANES 2005–2006, we prepared laboratory data following the same filter criteria as NHANES 2007–2008 and produced a final dataset of 2459 individuals, among which 637 (with retinopathy, 176; no retinopathy, 461) had diabetes.

### 3.2. Retinal Lesions of Diabetic Retinopathy

To help validate our methodology and cohort selection, we aimed to determine retinal lesions that define DR. To this end, we identified nine retinal lesions in NHANES 2007–2008 that were statistically significantly associated with DR and survived to *p*-value adjustment for false discovery (Table 2). The top lesions were retinal microaneurysms (*p* ≤ 0.0001), followed by retinal hard exudates (typically due to lipoprotein deposition in the retina and may be associated with macular oedema) (*p* ≤ 0.0001). Other key lesions at *p* ≤ 0.0001 were retinal soft exudate (now termed cotton wool spots), retinal blot haemorrhages, intraretinal microvascular abnormalities (IRMA), and macular oedema. In NHANES, retinal microaneurysms and retinal blot haemorrhages are encoded to be mutually exclusive, i.e., an individual is recorded as having retinal microaneurysms only when not accompanied with retinal blot haemorrhages, and vice-versa (Table 2).

### 3.3. Univariate Logistic Regression Analysis

In total, six variables reached statistical significance in the unadjusted univariate analysis, 11 after adjustment for age, two after adjustment for ethnicity, and seven for diabetes duration (Appendix A; Table 3). Glycohemoglobin (HbA1c) was the only variable that was statistically significant in both the unadjusted and adjusted analyses. Other risk variables of note that reached statistical significance in the unadjusted analysis included serum glucose (mmol/L) (i.e., RBS), osmolality (mmol/Kg), urinary albumin (mg/L), and fasting glucose (mmol/L) (i.e., FBS). The only protective variable, i.e., negatively associated, was haemoglobin (g/dL). These variables indicate suboptimal diabetes control, abnormal kidney function, and presence of anaemia as risk factors for DR. There was evidence of co-correlation among these statistically significant variables from the unadjusted analysis (Appendix A).

Interestingly, after adjustment for diabetes duration, the following variables reached statistical significance: HbA1c (%), osmolality (mmol/Kg), urinary iodine (µg/L), urinary cobalt (µg/L), urinary triclosan (ng/mL), urinary creatinine (µmol/L), and urinary barium (µg/L).

### 3.4. Principal Component Analysis

Unsupervised PCA using all laboratory variables revealed that 59 PCs could account for 80% or more variation in the dataset. Eighteen PCs were statistically significantly associated with DR at *p* ≤ 0.05 via independent binomial regression models testing each PC (Appendix A). The top variables responsible for variation along these PCs included measures of blood glucose (HbA1c, random blood glucose and fasting blood glucose), kidney function markers (urinary albumin, blood urea nitrogen (BUN)), haematological markers (haematocrit, haemoglobin, red blood cell distribution width), inflammatory markers (CRP), white blood cell count, urinary nitrates, segmented neutrophil count), and toxic elements (urinary beryllium and cotinine) among others—these PCs were also statistically significantly correlated to microaneurysms, the previously-identified top retinal lesion, and the covariates used during univariate testing (Appendix A). Through ROC analysis, these 18 PCs achieved an Area Under the Curve (AUC) of 0.796 (95% CI: 0.761–0.832).

### 3.5. Penalised Regression Model

We fitted an unbiased elastic-net penalised regression model to the laboratory variables and cross-validated it 100×. The model selected nine variables whose coefficients were not shrunk to zero: urinary albumin, BUN, urinary cobalt, CRP, HbA1c, blood osmolality, serum potassium, systolic blood pressure, and urinary nitrate (Figure 2). Of note, these measurements mainly represent diabetes and blood pressure control and kidney function. This model had an accuracy of 78.51% and AUC 0.74 (95% CI: 0.72–0.77) when predicted on the same dataset on which the model was produced.

### 3.6. RandomForest™ Classification Model

From our RandomForest™ model, HbA1c was the single best predictor of DR (mean decrease accuracy, 31.94%; Gini, 21.75) (Table 4). However, other notable variables of appreciable accuracy were markers of diabetes control (FBS, RBS) inflammation (CRP), kidney function (potassium, BUN, creatinine and urinary albumin), haematological markers (haematocrit), and systolic blood pressure, among others. The overall accuracy of the model on the validation cohort was 78.4% and AUC 0.71 (95% CI: 0.65–0.77).

### 3.7. Replication Cohort

In the NHANES 2005–2006 replication cohort, we performed penalised regression and RandomForest™ in the same way as per the 2007–2008 cohort. Our penalised regression model identified urinary albumin (mg/L), cockroach IgE antibody (kU/L), HbA1c (%), haemoglobin (g/dL), and urinary nitrate (ng/mL), with a model accuracy of 73.16% and AUC 0.76 (95% CI: 0.73–0.78). RandomForest™ identified HbA1c (%) as the variable contributing most to accuracy (mean decrease 16.96%), with many other variables contributing appreciable accuracy to the overall model (Appendix A)—the overall model accuracy was 72.98% and AUC 0.68 (95% CI: 0.61–0.75).

### 3.8. Final Clinical Risk Models

The 31 features identified by RandomForest™ (Table 4) were grouped into different categories of blood tests, according to diabetes status, haematocrit values, blood pressure (BP), immune markers, renal function, sterols, toxins and metals, and liver function. When modelled against DR outcome, each group varied in performance; diabetes tests alone achieved AUC 0.72 (95% CI: 0.68–0.76). A final clinical risk model comprising diabetes tests, BP, renal and liver function tests, haematocrit values, circulating sterols and immune markers achieved AUC 0.84 (95% CI: 0.78–0.9) (*p* = 0.00013) (Nagelkerke R^2^ 0.36) (Table 5; Figure 3).

## 4. Discussion

This EWAS of NHANES 2007–2008 data with DR outcomes in individuals with diabetes included an unbiased feature selection approach based on a rudimentary univariate regression enabled for parallel computation, PCA, penalised regression, and RandomForest™ of a large number of laboratory parameters.

In our rudimentary approach, which is ultimately running many univariate models in a parallelised fashion, HbA1c was the only variable to reach statistical significance after adjustment for age, ethnicity, and diabetes duration. The relationship between HbA1c and DR has been explored extensively, and was selected as the strongest risk factor in every approach we undertook, with a mean decrease accuracy of 31.94% via RandomForest™. On ranking all of the risk factors and available biomarkers, hyperglycaemia not only ranked the highest, but was about three times higher than the second highest on the rank, highlighting that, in clinical practice, the patient and all healthcare professionals should pay significant emphasis on optimal control of hyperglycaemia to prevent the development and progression of DR. Although it is well known that hyperglycaemia is a risk factor of DR, DR usually develops after over a decade of diabetes, and so, this study shows that all efforts should be made to avoid patient and clinician inertia in optimally controlling hyperglycaemia over the duration of diabetes, which remains a significant challenge in most patients.

The other traditional risk factor of hypertension also remained high in the rank, but was individually not as strong as hyperglycaemia. Both the penalised regression and RandomForest™ algorithms identified an association between elevated systolic blood pressure—but not diastolic—and DR (mean decrease accuracy, 1.9%), again confirming literature [10,28,29,30].

When we consider the top 10 risk factors or markers associated with DR, other than hyperglycaemia, inflammatory markers and renal function markers occupy the rest of the hierarchy in the RandomForest™. Although all models show that inflammatory markers, such as CRP and white blood cell counts, are as important as renal markers, systemic anti-inflammatory agents have not shown significant effect on DR. However, monitoring these markers and treating the inflammatory focus may prevent the development and progression of DR.

Further variables identified by both penalised regression and RandomForest™ were renal function tests including BUN, urinary albumin, potassium, osmolality, and urinary nitrate. These confirm the strong association of DR with markers of impaired kidney function.

Other known risk factors that contributed higher up in the ranking order include haematocrit (%) and cholesterol, and these are modifiable and may offer protection.

Although HbA1c has the strongest association with DR, our study highlight how the addition of other clinical parameters, e.g., from renal and liver function, and haematocrit can increase the sensitivity and specificity of predicting DR outcome, with our final clinical risk model achieving AUC 0.83 (95% CI: 0.77–0.89) (*p* = 0.00012) (Nagelkerke R^2^ 0.33), higher than any traditional diabetes control parameter in isolation or in combination.

The strengths of this study are that, in contrast to epidemiological studies that are typically conducted based on pre-conceived hypotheses and involve a single or just a few variables, the methods used in this study can be scaled to datasets of any size and therefore provide ways of working with large clinical and epidemiological datasets for the purpose of searching for novel hypotheses that could then lead to further focused investigations.

For example, the EWAS methodology and our RandomForest™ approach of non-targeted recursive feature selection also indicates a small contribution from toxins and metals, including 3-hydroxyphenanthrene, 9-hydroxyfluorene, phthalates, blood o-Xylene, and blood nitromethane. Therefore, retina may be a target tissue for environmental contamination. Some of the associations provide directions to future mechanistic research in DR. For example, we found that retinal microaneurysms (FDR-adjusted *p* ≤ 0.0001), the most statistically significant retinal lesions in individuals with DR, is already correlated with some of the variables such as HbA1c, CRP, BUN, beryllium, and haematocrit, suggesting early effects. In contrast, increased urinary cobalt, triclosan, and barium became significant only when adjusted for duration of diabetes. Most of these parameters are also linked to risk of allergies and lung disease, an association that has not been previously explored systematically.

We observed how the PCA-based model performed slightly better than others based on AUC, with 0.796 (95% CI: 0.761–0.832), compared to 0.74 (95% CI: 0.72–0.77) for penalised regression and 0.71 (95% CI: 0.65–0.77) for RandomForest™. Principal component analysis is a method that, via matrix calculations involving variance (or, more specifically, co-variance), transforms a dataset into uncorrelated PCs. Due to this uncorrelated nature and the fact that a large amount of total explained variation within a given dataset can be represented by a small number of PCs, this method works well for a complex dataset like the one in this current study. The penalised regression and RandomForest™ models gave comparable performance based on AUC, and even closer based on accuracy (78.51% vs. 78.4%, respectively). Unlike PCA, neither of these models are fundamentally based on variance (but can be indirectly influenced by it). Penalised regression with the lasso penalty is essentially a feature selection method based on logistic regression that aims to identify parameters with zero and non-zero effects; thus, the standardisation of the input parameters is important in order to mitigate effects of outliers. Penalised regression also expects that the boundary between the outcomes being measured is linear. RandomForest™, on the other hand, provides no inherent feature selection algorithm, instead assigning “importance” metrics to each parameter in the input model, metrics that can be used to set cut-offs for eliminating parameters and re-fitting the model. These models can also work with outcomes whose boundaries are non-linear in nature. It is therefore intuitive that penalised regression and RandomForest™ would result in similar performance metrics when both are correctly used. Equally, for a complex dataset, it is expected that they cannot supersede the performance of PCA.

As this is a cross-sectional study, a cause-effect relation cannot be established. Moreover, we are unable to rule out any confounding effects of any unmeasured factors. On the other hand, the main strength of the study is the use of the well characterised NHANES cohort in whom standardised protocols were used to measure laboratory parameters. We are not aware of any other association studies in DR where over 400 laboratory parameters were analysed simultaneously to develop multiple models. As the top variables of all four data driven agnostic models were similar, we also believe our findings are generalisable.

## 5. Conclusions

We confirm that DR is a complex disease and that the already established risk factors contribute significantly to the risk models of DR, with HbA1c being the strongest risk factor. Although our model provides an accuracy of approximately 80%, it also provides mechanistic insights into future research on DR including interrogating the interaction of low-ranking risk factors with more established factors in the models and highlights need to explore epigenetic screens to gauge better how risk factors influence gene expression. Most importantly, the study reinforces the need to control known risk factors of DR, especially hyperglycaemia.

## Figures and Tables

**Figure 1 jcm-09-03643-f001:**
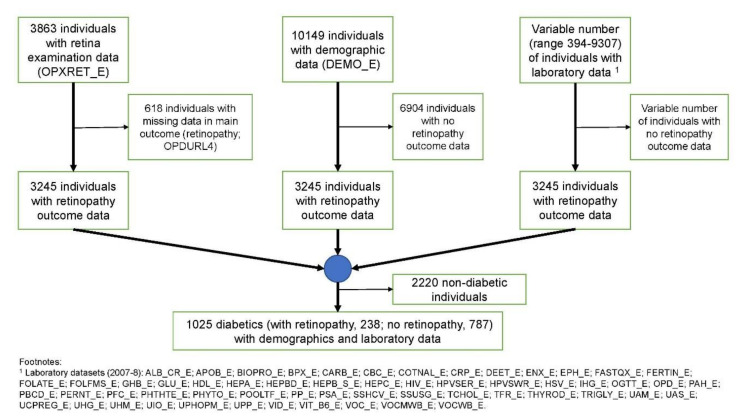
Cohort selection process for the National Health and Nutrition Examination Survey (NHANES) 2007–2008.

**Figure 2 jcm-09-03643-f002:**
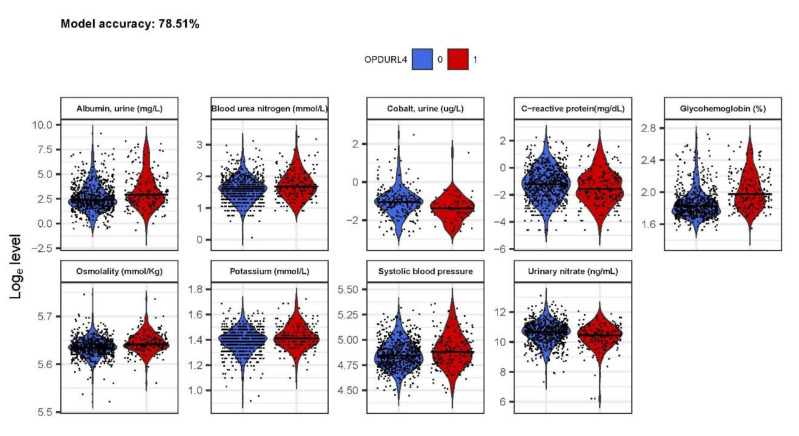
Penalised regression-selected variables. Variables were selected from a 100x cross-validated model with α = 0.5. Final variable selection was based on coefficients not shrunk to 0. Model accuracy was determined to be 78.4% accuracy and AUC 0.71 (95% CI: 0.65–0.77).

**Figure 3 jcm-09-03643-f003:**
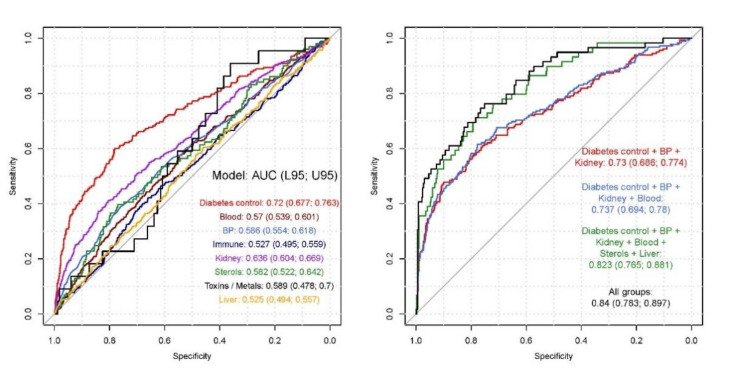
Final clinical risk models. Features from RandomForest™ were grouped logically based on similar function or clinical use (Table 4) and then tested independently in a univariate or multivariate regression model against diabetic retinopathy (DR) outcome. A final risk model including markers of hypertension, hypercholesterolemia, renal, and liver function tests, and haematocrit achieved AUC 0.84 (0.78–0.9).

**Table 1 jcm-09-03643-t001:** Demographic overview of study cohort.

Characteristics Mean (± SD) *n* (%)		Diabetes with Non-Diabetic Retinopathy (*n* = 787)	Diabetic Retinopathy (*n* = 238)	*p*-Value	β-Coefficient	OR (95% CI)
Age	-	62.39 (±11.02)	63.53 (±10.54)	NS	0.01	1.00 (1.00–1.02)
Sex	Male	420 (53.37)	126 (52.94)	-	-	-
	Female	367 (46.63)	112 (47.06)	NS	0.02	1.02 (0.76–1.36)
Ethnicity	Non-Hispanic white	369 (46.88)	91 (38.23)	-	-	-
	Non-Hispanic black	166 (21.09)	74 (31.09)	**	0.59	1.81 (1.27–2.58)
	Mexican-American	138 (17.54)	41 (17.23)	NS	0.19	1.21 (0.79–1.83)
	Other Hispanic	89 (11.31)	28 (11.77)	NS	0.24	1.28 (0.79–2.07)
	Other race—including multiracial	25 (3.18)	4 (1.68)	NS	−0.43	0.65 (0.22–1.91)
Education	Less than 9th grade	148 (18.8)	54 (22.69)	-	-	-
	9–11th grade ^±^	148 (18.8)	51 (21.43)	NS	−0.06	0.94 (0.61–1.47)
	High school graduate/GED or equivalent	199 (25.29)	56 (23.53)	NS	−0.26	0.77 (0.5–1.19)
	Some college or AA degree	172 (21.86)	55 (23.11)	NS	−0.13	0.88 (0.57–1.35)
	College graduate or above	120 (15.25)	22 (9.24)	*	−0.69	0.5 0.29–0.87)
Marital status	Married	464 (58.96)	133 (55.88)	-	-	-
	Widowed	119 (15.12)	36 (15.13)	NS	0.05	1.06 (0.69–1.61)
	Divorced	90 (11.44)	37 (15.55)	NS	0.36	1.43 (0.94–2.2)
	Separated	32 (4.07)	7 (2.94)	NS	−0.27	0.76 (0.33–1.77)
	Never married	53 (6.73)	18 (7.56)	NS	0.17	1.19 (0.67–2.09)
	Living with partner	29 (3.68)	7 (2.94)	NS	−0.17	0.84 (0.36–1.97)
Family income	$0–$4999	11 (1.4)	3 (1.26)	-	-	-
	$5000–$9999	47 (5.97)	12 (5.04)	NS	−0.07	0.94 (0.23–3.89)
	$10,000–$14,999	82 (10.42)	22 (9.25)	NS	−0.02	0.98 (0.25–3.84)
	$15,000–$19,999	68 (8.64)	22 (9.25)	NS	0.17	1.19 (0.30–4.64)
	$20,000–$24,999	69 (8.77)	25 (10.5)	NS	0.28	1.33 (0.34–5.16)
	$25,000–$34,999	103 (13.09)	42 (17.65)	NS	0.40	1.50 (0.40–5.63)
	$35,000–$44,999	68 (8.64)	23 (9.66)	NS	0.22	1.24 (0.32–4.84)
	$45,000–$54,999	55 (6.99)	12 (5.04)	NS	−0.22	0.80 (0.19–3.31)
	$55,000–$64,999	41 (5.21)	15 (6.3)	NS	0.29	1.34 (0.33–5.48)
	$65,000–$74,999	36 (4.57)	6 (2.52)	NS	−0.49	0.61 (0.13–2.86)
	$75,000–$99,999	44 (5.59)	12 (5.04)	NS	0.00	1.00 (0.24–4.17)
	≥$100,000	88 (11.18)	17 (7.14)	NS	−0.35	0.71 (0.18–2.81)
	Over $20,000	31 (3.94)	9 (3.78)	NS	0.06	1.06 (0.24–4.66)
	Under $20,000	17 (2.16)	2 (0.84)	NS	−0.84	0.43 (0.06–3.01)
	Missing	27 (3.43)	16 (6.73)	-	-	-
Diabetes duration	-	9.05 (±11.05)	16.33 (±12.57)	***	0.05	1.05 (1.04–1.07)

Notes: The NHANES codes used in our selection criteria of people with diabetes include: 1, self-reported diabetes (DIQ010); 2, on anti-diabetes drugs (DIQ070); 3, taking insulin (DIQ050); 4, fasting blood sugar (FBS) ≥ 6.1 (110 mg/dl) (LBDGLUSI); 5, random blood sugar (RBS) ≥ 11.1 (200 mg/dl) (LBDSGLSI); 6, oral glucose tolerance test (OGTT) ≥ 200 mg/dl (LBDGLTSI); 7, glycohemoglobin (HbA1c) ≥ 6.5% (LBXGH). Ethnicity, education, and diabetes duration contain at least one term that is statistically significant. ± includes 12th grade with no diploma. NS, not statistically significant; *, *p* < 0.05; **, *p* < 0.01; ***, *p* < 0.001.

**Table 2 jcm-09-03643-t002:** Retinal lesions that constitute diabetic retinopathy outcome.

Clinical Signs of DR	*n* (%)	β-Coefficient	OR (95% CI)	*p*-Value	FDR-Adjusted *p*-Value
Retinal microaneurysms only, worse eye	129 (12.59)	5.67	288.87 (127.66–653.68)	***	***
Retinal hard exudate, worse eye	86 (8.39)	3.72	41.34 (20.31–84.17)	***	***
Retinal blot haemorrhages, worse eye	47 (4.59)	3.36	28.71 (14.11–58.42)	***	***
Retinal soft exudate, worse eye	76 (7.41)	4.2	66.49 (26.44–167.21)	***	***
IRMA, worse eye	62 (6.05)	2.85	17.36 (9.06–33.26)	***	***
Macular oedema, worse eye	51 (4.98)	3.88	48.24 (17.18–135.44)	***	***
Retinal fibrous proliferation, worse eye	19 (1.85)	3.41	30.19 (6.92–131.67)	***	***
Macular oedema in centre, worse eye	26 (2.54)	4.53	92.33 (12.45–684.9)	***	***
Retinal new vessels elsewhere, worse eye	15 (1.46)	3.12	22.68 (5.08–101.23)	***	***

Notes: lesions are taken from the NHANES ophthalmology—retinal imaging (OPXRET_E) dataset. Only lesions with false discovery rate (FDR)-adjusted *p* ≤ 0.05 are listed. Soft exudate is now termed cotton wool spots. NS, not statistically significant; IRMA, intraretinal microvascular abnormalities; ***, *p* < 0.001.

**Table 3 jcm-09-03643-t003:** Laboratory variables associated with retinopathy in individuals with diabetes.

	Unadjusted/Non-Covariate Adjusted				Age-Adjusted				Ethnicity-Adjusted				Diabetes Duration-Adjusted			
Description	β-Coefficient	OR (95% CI)	*p*-Value	FDR-Adjusted *p*-Value	β-Coefficient	OR (95% CI)	*p*-Value	FDR-Adjusted *p*-Value	β-Coefficient	OR (95% CI)	*p*-Value	FDR-Adjusted *p*-Value	β-Coefficient	OR (95% CI)	*p*-Value	FDR-Adjusted *p*-Value
Glycohemoglobin (HbA1c) (%)	0.82	2.27 (1.84–2.8)	***	***	0.85	2.34 (1.87–2.92)	***	**	0.83	2.28 (1.82–2.87)	***	**	0.72	2.05 (1.55–2.73)	***	*
Random Glucose, serum (RBS) (mmol/L)	0.54	1.72 (1.42–2.08)	***	**	0.57	1.77 (1.45–2.15)	***	*	0.54	1.71 (1.41–2.09)	***	*	0.36	1.43 (1.1–1.86)	*	NS
Osmolality (mmol/Kg)	0.49	1.63 (1.34–1.99)	***	*	0.45	1.57 (1.3–1.9)	***	*	0.49	1.63 (1.34–1.99)	***	NS	0.39	1.48 (1.12–1.94)	*	*
Albumin, urine (mg/L)	0.45	1.57 (1.28–1.93)	***	*	0.43	1.53 (1.24–1.88)	**	*	0.42	1.53 (1.25–1.87)	**	NS	0.25	1.28 (0.98–1.68)	NS	NS
Haemoglobin (g/dL)	−0.33	0.72 (0.61–0.85)	**	*	−0.30	0.74 (0.63–0.88)	**	*	−0.29	0.75 (0.62–0.9)	**	NS	0.00	1 (0.73–1.37)	NS	NS
Fasting Glucose(FBS) (mmol/L)	0.49	1.63 (1.28–2.07)	**	*	0.51	1.66 (1.3–2.13)	**	*	0.46	1.59 (1.25–2.02)	**	NS	0.22	1.24 (0.93–1.66)	NS	NS
4-(methylnitrosamino)-1-(3-pyridyl)-1-butanol (NNAL), urine (ng/mL)	−0.26	0.77 (0.67–0.88)	**	NS	−0.20	0.82 (0.71–0.94)	*	NS	−0.28	0.75 (0.65–0.87)	**	NS	−0.25	0.78 (0.52–1.19)	NS	NS
Iodine, urine (ug/L)	−0.17	0.84 (0.76–0.93)	**	NS	−0.21	0.81 (0.73–0.9)	**	*	−0.15	0.86 (0.77–0.96)	*	NS	−0.29	0.75 (0.63–0.89)	**	*
Cobalt, urine (ug/L)	−0.51	0.6 (0.44–0.82)	**	NS	−0.50	0.6 (0.45–0.82)	**	*	−0.51	0.6 (0.44–0.81)	**	NS	−0.52	0.59 (0.45–0.78)	**	*
Haematocrit (%)	−0.31	0.73 (0.6–0.89)	**	NS	−0.28	0.76 (0.62–0.92)	*	NS	−0.28	0.75 (0.62–0.92)	*	NS	−0.03	0.97 (0.7–1.35)	NS	NS
Blood urea nitrogen (BUN) (mmol/L)	0.33	1.4 (1.13–1.73)	**	NS	0.27	1.31 (1.01–1.71)	NS	NS	0.35	1.43 (1.16–1.75)	**	NS	0.15	1.17 (0.87–1.57)	NS	NS
Albumin (g/L)	−0.22	0.8 (0.69–0.93)	*	NS	−0.21	0.81 (0.69–0.94)	*	NS	−0.19	0.83 (0.7–0.99)	NS	NS	−0.09	0.92 (0.74–1.14)	NS	NS
Urinary Triclosan (ng/mL)	−0.42	0.65 (0.49–0.88)	*	NS	−0.40	0.67 (0.5–0.89)	*	NS	−0.42	0.66 (0.49–0.89)	*	NS	−0.60	0.55 (0.36–0.83)	*	*
Mean cell haemoglobin (pg)	−0.24	0.79 (0.66–0.93)	*	NS	−0.27	0.77 (0.65–0.91)	**	*	−0.17	0.84 (0.7–1.02)	NS	NS	−0.01	0.99 (0.76–1.28)	NS	NS
Lead, urine (µg/L)	−0.40	0.67 (0.49–0.91)	*	NS	−0.40	0.67 (0.49–0.91)	*	NS	−0.41	0.66 (0.49–0.9)	*	NS	−0.43	0.65 (0.43–0.99)	NS	NS
Creatinine, urine (µmol/L)	−0.22	0.8 (0.67–0.97)	*	NS	−0.19	0.83 (0.68–1)	NS	NS	−0.27	0.77 (0.64–0.92)	*	NS	−0.28	0.76 (0.64–0.9)	**	*
Alanine aminotransferase (ALT) (U/L)	−0.25	0.78 (0.62–0.97)	*	NS	−0.20	0.82 (0.66–1.02)	NS	NS	−0.22	0.81 (0.64–1.01)	NS	NS	−0.09	0.92 (0.68–1.23)	NS	NS
Creatinine (µmol/L)	0.24	1.27 (1.04–1.54)	*	NS	0.18	1.19 (0.96–1.48)	NS	NS	0.20	1.23 (0.99–1.53)	NS	NS	0.13	1.14 (0.89–1.45)	NS	NS
Red blood cell count (million cells/µL)	−0.19	0.82 (0.7–0.97)	*	NS	−0.13	0.88 (0.74–1.04)	NS	NS	−0.19	0.83 (0.7–0.98)	NS	NS	0.01	1.01 (0.74–1.37)	NS	NS
Mean cell volume (fL)	−0.21	0.81 (0.68–0.98)	*	NS	−0.25	0.78 (0.65–0.93)	*	NS	−0.15	0.86 (0.71–1.06)	NS	NS	−0.05	0.95 (0.72–1.26)	NS	NS
Platelet count (1000 cells/µL)	−0.21	0.81 (0.68–0.98)	*	NS	−0.18	0.83 (0.68–1.02)	NS	NS	−0.23	0.79 (0.67–0.94)	*	NS	−0.29	0.75 (0.57–0.99)	NS	NS
Mean platelet volume (fL)	0.24	1.27 (1.04–1.55)	*	NS	0.26	1.3 (1.06–1.59)	*	NS	0.22	1.25 (1.02–1.53)	NS	NS	0.09	1.09 (0.83–1.43)	NS	NS
Cotinine (ng/mL)	−0.15	0.86 (0.76–0.99)	*	NS	−0.09	0.91 (0.79–1.04)	NS	NS	−0.18	0.84 (0.74–0.95)	*	NS	−0.06	0.94 (0.66–1.34)	NS	NS
Insulin (pmol/L)	−0.32	0.73 (0.55–0.97)	*	NS	−0.30	0.74 (0.55–1)	NS	NS	−0.29	0.75 (0.56–1)	NS	NS	−0.22	0.8 (0.51–1.26)	NS	NS
Blood cadmium (nmol/L)	−0.23	0.8 (0.65–0.97)	*	NS	−0.25	0.78 (0.64–0.95)	*	NS	−0.23	0.79 (0.64–0.98)	NS	NS	−0.28	0.76 (0.5–1.16)	NS	NS
Urinary perchlorate (ng/mL)	−0.25	0.78 (0.63–0.95)	*	NS	−0.24	0.78 (0.63–0.98)	*	NS	−0.24	0.78 (0.64–0.96)	*	NS	−0.31	0.73 (0.53–1.01)	NS	NS
Urinary nitrate (ng/mL)	−0.28	0.75 (0.6–0.94)	*	NS	−0.23	0.79 (0.63–1)	NS	NS	−0.27	0.76 (0.6–0.96)	*	NS	−0.34	0.71 (0.55–0.91)	*	NS
Cesium, urine (µg/L)	−0.33	0.72 (0.54–0.95)	*	NS	−0.32	0.72 (0.55–0.96)	*	NS	−0.33	0.72 (0.54–0.95)	*	NS	−0.34	0.71 (0.47–1.06)	NS	NS
Thallium, urine (µg/L)	−0.40	0.67 (0.48–0.95)	*	NS	−0.39	0.68 (0.48–0.96)	*	NS	−0.42	0.66 (0.47–0.92)	*	NS	−0.48	0.62 (0.4–0.94)	*	NS
Vitamin D2 or D3 25OHD2 + 25OHD3 (nmol/L)	−0.24	0.79 (0.65–0.95)	*	NS	−0.25	0.78 (0.65–0.94)	*	NS	−0.18	0.83 (0.66–1.04)	NS	NS	−0.20	0.82 (0.61–1.1)	NS	NS
Blood Toluene (ng/mL)	−0.21	0.81 (0.68–0.96)	*	NS	−0.20	0.82 (0.69–0.97)	*	NS	−0.22	0.8 (0.65–0.98)	NS	NS	−0.30	0.74 (0.45–1.21)	NS	NS
C-reactive protein (mg/dL)	−0.19	0.82 (0.69–0.99)	NS	NS	−0.18	0.83 (0.69–1.01)	NS	NS	−0.25	0.78 (0.66–0.92)	*	NS	−0.27	0.77 (0.6–0.98)	*	NS
Barium, urine (µg/L)	−0.39	0.68 (0.47–0.99)	NS	NS	−0.38	0.69 (0.48–0.99)	NS	NS	−0.38	0.68 (0.47–1)	NS	NS	−0.44	0.64 (0.47–0.87)	*	*
Urinary 4-tert-octylphenol (ng/mL)	−0.34	0.71 (0.45–1.12)	NS	NS	−0.33	0.72 (0.46–1.12)	NS	NS	−0.42	0.66 (0.43–1.01)	NS	NS	−0.82	0.44 (0.23–0.87)	*	NS
Dimethyl dithiophosphate (µg/L)	−0.31	0.74 (0.51–1.06)	NS	NS	−0.35	0.71 (0.49–1.01)	NS	NS	−0.21	0.81 (0.59–1.12)	NS	NS	−0.67	0.51 (0.29–0.91)	*	NS

Variables were first tested in an unadjusted/non-covariate adjusted analysis, and then again adjusting for age, ethnicity, and diabetes duration. To provide a broad overview, any variable passing nominal (i.e., prior to FDR-correction) *p* ≤ 0.05 from either the non-covariate adjusted or any of the covariate-adjusted analyses are listed. OR, odds ratio; NS, not statistically significant; *, *p* < 0.05; **, *p* < 0.01; ***, *p* < 0.001.

**Table 4 jcm-09-03643-t004:** RandomForest™-selected variables (features) ranked by mean decrease accuracy.

Marker	Group	Mean Decrease Accuracy	Mean Decrease Gini
Glycohemoglobin (%)	Diabetes status	31.93719324	21.75225602
C-reactive protein (mg/dL)	Immune markers	11.51047187	10.1975135
Potassium (mmol/L)	Renal function	11.44142126	8.198116194
Albumin, urine (mg/L)	Renal function	8.220187056	10.21346621
Monocyte number (1000 cells/uL)	Immune markers	7.663589246	3.957711466
Osmolality (mmol/Kg)	Renal function	7.510989556	5.33445026
White blood cell count (1000 cells/uL)	Immune markers	7.440157644	4.405168072
Blood urea nitrogen (mmol/L)	Renal function	7.224789174	5.073947519
Segmented neutrophils num (1000 cell/uL)	Immune markers	7.020397225	4.067402271
Fasting Glucose (mmol/L)	Diabetes status	6.563694988	2.826827797
Red cell distribution width (%)	Haematocrit	6.138538515	4.947185792
Urinary nitrate (ng/mL)	Renal function	5.899174386	5.844258103
Glucose, serum (mmol/L)	Diabetes status	5.85339684	4.77357191
2-hydroxyphenanthrene (ng/L)	Toxins/Metals	4.028082549	2.307374348
MCHC (g/dL)	Haematocrit	3.936015913	3.896804592
Creatinine (µmol/L)	Renal function	3.530916132	3.747500758
Mono-2-ethyl-5-carboxypentyl phthalate	Toxins/Metals	2.996581682	2.212151134
Blood Nitromethane (pg/mL)	Toxins/Metals	2.936160917	3.31634662
Phosphorus (mmol/L)	Toxins/Metals	2.819084205	3.768671211
Total Cholesterol (mmol/L)	Sterols	2.448578413	3.833301666
Enterodiol (ng/mL)	Sterols	2.401651721	2.722762228
Haematocrit (%)	Haematocrit	2.364741874	4.841281382
Mono-n-octyl phthalate (ng/mL)	Toxins/Metals	2.215508752	0.231647322
Mean cell haemoglobin (pg)	Haematocrit	2.183482824	4.292282119
Gamma glutamyl transferase (U/L)	Liver Function	1.989695745	4.015070221
Systolic blood pressure	Blood pressure	1.892815461	8.85430752
Blood o-Xylene (ng/mL)	Toxins/Metals	1.670964308	3.708248225
Lactate dehydrogenase LDH (U/L)	Liver Function	1.593869272	4.231659273
9-hydroxyfluorene (ng/L)	Toxins/Metals	1.430814333	2.206778568
Cholesterol (mmol/L)	Sterols	1.201366974	4.025207689
3-hydroxyphenanthrene (ng/L)	Toxins/Metals	1.00569817	1.63630888

Notes: The model was initially trained on all laboratory variables in an unsupervised fashion, with Kappa-based model tuning to select the optimum values for ‘mtry’ (the ideal number of variables to randomly sample) and ‘ntrees’ (the ideal number of trees). Only variables contributing >1% mean decrease in accuracy from the initial model were retained, followed by recursive steps to remove low-informative variables. Variables are manually assigned to groups based on similar organ function or other characteristic. Mean decrease accuracy is the estimated decrease in overall model accuracy once a given parameter is removed from the model, and therefore serves as a useful metric for ordering importance. The Gini importance measure relates to the ‘splitting’ criterion that is employed in classification trees, and it is known to be less biased for continuous variables (26), which naturally have more splitting points compared to categorical variables. ‘Group’ is manually curated.

**Table 5 jcm-09-03643-t005:** Final clinical risk models.

Model	Wald Test *p*-Value	McFadden R^2^	Nagelkerke R^2^	AUC (95% CI)
Diabetes Status	***	0.102	0.142	0.72 (0.677–0.763)
Haematocrit	**	0.007	0.009	0.57 (0.539–0.601)
Blood Pressure (BP)	***	0.017	0.024	0.586 (0.554–0.618)
Immune Markers	NS	0.002	0.002	0.527 (0.495–0.559)
Renal function tests (renal)	***	0.039	0.054	0.636 (0.604–0.669)
Sterols (include cholesterol)	NS	0.012	0.017	0.582 (0.522–0.642)
Toxins/Metals	NS	0.029	0.041	0.589 (0.478–0.7)
Liver function tests	NS	0.002	0.003	0.525 (0.494–0.557)
Diabetes control + BP + Renal function	***	0.13	0.18	0.73 (0.686–0.774)
Diabetes control + BP + renal function + Haematocrit	***	0.135	0.184	0.737 (0.694–0.78)
Diabetes control + BP + renal function + Haematocrit + Sterols + Liver function	***	0.238	0.315	0.823 (0.765–0.881)
All groups ±	***	0.272	0.355	0.84 (0.783–0.897)

Features from RandomForest™ were grouped logically based on similar function or clinical use (Table 4) and then tested independently in a univariate or multivariate regression model against DR outcome. ± The only toxins/metal included was phosphorus (mmol/L)—others filtered out due to high missingness (>50%), resulting in difficulty fitting model. NS, not statistically significant; **, *p* < 0.01; ***, *p* < 0.001.

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
