# Peer review of "Diabetic Retinopathy Environment-Wide Association Study (EWAS) in NHANES 2005–2008"

_jcm, 2020, doi:10.3390/jcm9113643_

Round 1
Reviewer 1 Report
The study of Blighe et al shows that, after performing an environment-wide association study (EWAS) of laboratory parameters available in National Health and Nutrition Examination Survey (NHANES) 2007-8 in individuals with diabetes and diabetic retinopathy (DR) and after replicating the models in NHANES 2005-6, it was possible to demonstrate that, among established and novel biomarkers that have been associated to the risk to develop DR, the established ones, and in particular hyperglycemia, still remain as the major risk factors to develop DR.
Comments to the manuscript are:
1) This is a well designed and well performed study that includes data coming from a large number of patients affected by diabetes
2) The diagnosis of DR is based on evaluation of retinal imaging. Were the images based on 7-field fundus photography?
Author Response
1) This is a well designed and well performed study that includes data coming from a large number of patients affected by diabetes
We thank the reviewer for the comment.
2) The diagnosis of DR is based on evaluation of retinal imaging. Were the images based on 7-field fundus photography?
It is 4-field.
Reviewer 2 Report
Summary:
- The author stated that the association between circulating biomarkers and known DR risk factors remains unclear but do not sufficiently differentiate between the circulating biomarkers and the DR risk factors? In the main body of the study, all factors seem to become “markers” and are analysed together.
- Research in context: Not sure about the presentation using bullet point and leading questions?
- 2nd bullet point: what is meant by “the variance of this complication”?
- 4th bullet point: As no biomarkers are reported yet, just say “There is an unmet need ……to help rank their associations with DR.”
- 5th,6th bullet point: avoid using “We”
- Keywords are missing.
- Methods should list the mathematical models used.
Introduction
- Page 2 first sentence: The authors state that “Diabetes represents the most common cause of microvascular changes in the retina.” Diabetic retinopathy (DR), a chronic diabetic complication is generally believed to be the cause of the microvascular changes in the retina, not diabetes. Could the authors therefore rectify this sentence accordingly?
- Avoid use of symbols such as >, ~
- The structure of introduction does not flow well. Introduction to the Risk factors, NHANES, DR grading and DME should be in their own separate paragraphs.
- The authors do not sufficiently convey why they think it is important to “rank” the biomarkers. Could this be because some of these biomarkers correlate with DR progression more strongly than others?
- The historical details of NHANES provided by the authors is not necessary.
- The authors stated that their aim was “to evaluate the rank order of systemic and laboratory risks, of DR among individuals identified as having diabetes to evaluate their relative associations with DR”. However, it is not clear what the difference between systemic and laboratory risks are.
Methods
- The characteristics assessed should be listed: age, sex, etc.
- The column width of Table 1. needs to be spaced more evenly. The first 2 columns and the heading of the third column should take up less space.
- The selection criteria needs to be clearly stated under methods, not as a separate note under Table 1.
- Headings should be more descriptive.
- The capital codes e.g. fasting blood sugar (LBDGLUSI) interrupts the flow of the writing – where are those codes from? Remove if they do not add further details to this study, include them in a separate table if they are relevant and explain their use.
- The source of funding should be moved to the appropriate section at the end of the manuscript.
Results
- I was unable to see figures 1, 2, and 3 as well as the supplementary figure. Could the authors please ensure that this is included as it was difficult to understand the manuscript without them.
- A general description for each model’s significance should be stated for the general reader to understand.
- For all tables, please describe any abbreviations following the full name of the marker e.g. Glycohemoglobin (HbA1c). Otherwise use the abbreviation in the table and then write out in full in the legends of the tables.
- Use * to indicate statistical significance because it was very hard to find them as there are so many markers being analysed.
- Table 2: the heading “retinal co-morbidity” seems incorrect. Retinopathy, nephropathy and neuropathy are co-morbidities of diabetes. The heading should be changed to ‘clinical signs of DR’.
- Table 4: title should indicate that the markers are “ranked” by mean decrease accuracy. Could the authors explain what they mean by the “mean decrease accuracy”? If not important then please group them according to their “Group”. The information about the mean decrease accuracy should also be included in the methods section.
Discussion:
- There was no clear explanation of how the study addressed their aim/hypothesis.
- It will be useful to analyse the correlation between HbA1c and different DR severity levels, using the grading scale International Clinical Diabetic Retinopathy Disease Severity Scale. Based on their findings, could the authors comment on this?
- The authors analysed DR signs but did not explain their findings with the rest of the data. Could the authors include a discussion of their DR findings?
Author Response
Summary
The author stated that the association between circulating biomarkers and known DR risk factors remains unclear but do not sufficiently differentiate between the circulating biomarkers and the DR risk factors? In the main body of the study, all factors seem to become “markers” and are analysed together.
These words are used interchangeably and depend on the field in which one has worked, e.g., statistics versus programming versus bioinformatics. We have relabelled these throughout the text in an attempt to make the naming more consistent.
Research in context: Not sure about the presentation using bullet point and leading questions?
We have changed this to be a standard paragraph of text
2nd bullet point: what is meant by “the variance of this complication”?
We have changed this to 'explain the presence of DR in people with diabetes' (line 34)
4th bullet point: As no biomarkers are reported yet, just say “There is an unmet need ……to help rank their associations with DR.”
We have modified this (line 38)
5th,6th bullet point: avoid using “We”
Done
Keywords are missing.
These are now added (line 69)
Methods should list the mathematical models used.
The methods currently go into detail about how any user can reproduce the analysis via coding, including details on the exact parameters to use for each function.
Introduction
Page 2 first sentence: The authors state that “Diabetes represents the most common cause of microvascular changes in the retina.” Diabetic retinopathy (DR), a chronic diabetic complication is generally believed to be the cause of the microvascular changes in the retina, not diabetes. Could the authors therefore rectify this sentence accordingly?
We have now rectified this (line 72)
Avoid use of symbols such as >, ~
We have changed these where relevant
The structure of introduction does not flow well. Introduction to the Risk factors, NHANES, DR grading and DME should be in their own separate paragraphs.
Done - the introduction has been extensively modified
The authors do not sufficiently convey why they think it is important to “rank” the biomarkers. Could this be because some of these biomarkers correlate with DR progression more strongly than others?
We have added a new paragraph from line 93 in relation to this.
The historical details of NHANES provided by the authors is not necessary.
We have shortened this somewhat but believe that a very brief overview of the study is still important for readership that may not be aware of it. It is anticipated that some readers outside of epidemiology may read the work for, e.g., the RandomForest and EWAS methodologies. From line 99.
The authors stated that their aim was “to evaluate the rank order of systemic and laboratory risks, of DR among individuals identified as having diabetes to evaluate their relative associations with DR”. However, it is not clear what the difference between systemic and laboratory risks are.
We added a new paragraph from line 93 in relation to this
Methods
The characteristics assessed should be listed: age, sex, etc.
These are added on line 230
The column width of Table 1. needs to be spaced more evenly. The first 2 columns and the heading of the third column should take up less space.
Done
The selection criteria needs to be clearly stated under methods, not as a separate note under Table 1.
Done - from line 233
Headings should be more descriptive.
These have been modified
The capital codes e.g. fasting blood sugar (LBDGLUSI) interrupts the flow of the writing – where are those codes from? Remove if they do not add further details to this study, include them in a separate table if they are relevant and explain their use.
We have removed these
The source of funding should be moved to the appropriate section at the end of the manuscript.
We have moved this to Acknowledgments
Results
I was unable to see figures 1, 2, and 3 as well as the supplementary figure. Could the authors please ensure that this is included as it was difficult to understand the manuscript without them. Please submit.
We are not sure what happened - the Figures were uploaded in the submission system and the Supplementary Data was also uploaded. We will pass this message to the editor(s).
A general description for each model’s significance should be stated for the general reader to understand.
We have added a new paragraph to Discussion that attempts to explain these. Starting from 'We observed how the PCA-based model performed slightly...'
For all tables, please describe any abbreviations following the full name of the marker e.g. Glycohemoglobin (HbA1c). Otherwise use the abbreviation in the table and then write out in full in the legends of the tables.
Done
Use * to indicate statistical significance because it was very hard to find them as there are so many markers being analysed.
We have changed these in all Tables in the main text, and adjusted the Table notes to reflect these.
Table 2: the heading “retinal co-morbidity” seems incorrect. Retinopathy, nephropathy and neuropathy are co-morbidities of diabetes. The heading should be changed to ‘clinical signs of DR’.
Done
Table 4: title should indicate that the markers are “ranked” by mean decrease accuracy. Could the authors explain what they mean by the “mean decrease accuracy”? If not important then please group them according to their “Group”. The information about the mean decrease accuracy should also be included in the methods section.
We have added a short explanation of mean decrease accuracy to the Note below Table 4. We hope that this is sufficient?
Reviewer 3 Report
This is rather statistical-based study. The major finding is that hyperglicemia
is the strongest risk factor for diabetic retinopathy, that is the confirmtion of previous studies. The discussion section is very short and does not add anyhting important to the knowledge. The scientific significance of this study is low in my opinion.
Author Response
This is rather statistical-based study. The major finding is that hyperglicemia is the strongest risk factor for diabetic retinopathy, that is the confirmtion of previous studies. The discussion section is very short and does not add anyhting important to the knowledge. The scientific significance of this study is low in my opinion.
We thank the reviewer for taking time to review our work. We have greatly expanded the Discussion section in the hope that this changes the author's opinion.
Round 2
Reviewer 3 Report
The manuscript has been improved according to sugegstions.Discussion is extended, the significance of findings is better described.